# Prevalence and identification of anxiety disorders in pregnancy: the diagnostic accuracy of the two-item Generalised Anxiety Disorder scale (GAD-2)

Selina Nath,[1] Elizabeth G Ryan,[2,3] Kylee Trevillion,[1] Debra Bick,[4] Jill Demilew,[5] Jeannette Milgrom,[6] Andrew Pickles,[2] Louise M Howard[1,4,7]

For numbered affiliations see end of article.

**Correspondence to**
Dr Selina Nath;
selina.nath@kcl.ac.uk

## ABSTRACT

**Objective** To estimate the population prevalence of anxiety disorders during pregnancy and investigate the diagnostic accuracy of the two-item Generalised Anxiety Disorder scale (GAD-2) for a) GAD and b) any anxiety disorder.

**Design** Cross-sectional survey using a stratified sampling design. Sampling weights were used in the analysis to adjust for the bias introduced by the stratified sampling.

**Setting** Inner-city maternity service, South London.

**Participants** 545 pregnant women were interviewed after their first antenatal appointment; 528 provided answers on the GAD-2 questions.

**Main outcome measures** Diagnosis generated by the Structured Clinical Interview for Diagnostic and Statistical Manual of Mental Disorders, 4th edition (SCID).

**Results** Population prevalence of anxiety disorders was 17% (95% CI 12% to 21%): 5% (95% CI 3% to 6%) for GAD, 4% (95% CI 2% to 6%) for social phobia, 8% (95% CI 5% to 11%) for specific phobia and 2% (95% CI 1% to 4%) for obsessive-compulsive disorder. Post-traumatic stress disorder (PTSD) prevalence was unclear due to higher levels of reluctance to respond to PTSD interview questions but sensitivity analyses suggest population prevalence maybe up to 4% (95% CI 2% to 6%). Weighted sensitivity of GAD-2 for GAD (cut-off ≥3) was 69%, specificity 91%, positive predictive value 26%, negative predictive value 98% and likelihood ratio 7.35. For any anxiety disorder the weighted sensitivity was 26%, specificity 91%, positive predictive value 36%, negative predictive value 87% and likelihood ratio 2.92.

**Conclusions** Anxiety disorders are common but GAD-2 generates many false positives and may therefore be unhelpful in maternity services.

## INTRODUCTION

Anxiety disorders are more common in women than men,[1 2] and the perinatal period (ie, pregnancy and the year after birth) has been reported as a particularly vulnerable time for the onset or relapse of anxiety disorders in women.[3 4] Antenatal anxiety disorders have been associated with adverse pregnancy outcomes, including preterm birth, low birth weight, lower Apgar scores, postpartum anxiety and depression and adverse child developmental outcomes[5–7] including difficult temperament, increased sleep problems, bonding/attachment problems and poorer emotional, behavioural and cognitive development.[5–10] A recent systematic review and meta-analysis reported a pooled prevalence of 4% (95% CI 2% to 6%; 10 studies with pooled n=6910) for Generalised Anxiety Disorder (GAD) and 15% (95% CI 9% to 21%; 9 studies with pooled n=4648) for any anxiety disorders during pregnancy based on studies conducted outside of the UK where diagnostic clinical interviews were used, and 23% when using cut-offs on validated self-report questionnaires.[11] Anxiety disorders are treatable, so early detection and treatment during the antenatal period, when women are in regular contact with healthcare professionals could prevent adverse outcomes.[12–14] However, despite regular contact with healthcare professionals during pregnancy, antenatal mental disorders are often undetected and untreated.[15 16]

The National Institute for Health and Care Excellence (NICE) (CG192; 2014) suggested that maternity professionals could consider the use of the two-item GAD tool (GAD-2) to identify anxiety disorders during pregnancy and after birth, although also highlighted the lack of evidence on the use of the GAD-2 in early pregnancy. The recommendation was therefore driven by concern about the high prevalence of anxiety disorders (NICE 2014).[12 17] This extends the focus of early intervention from a previous emphasis on identification of perinatal depression to other perinatal mental disorders, including comorbid conditions. Outside of the perinatal period, a systematic review and meta-analysis[18] of studies in men and women reported that the GAD-2 (using a cut-off of ≥3) showed fairly high pooled sensitivity of 0.76 (95% CI 0.55 to 0.89), and pooled specificity of 0.81 (95% CI 0.60 to 0.92) for GAD (five studies with pooled n=1987). For detecting any anxiety disorder, the systematic review[18] reported a moderate sensitivity (range: 0.65–0.72) and unclear specificity (range: 0.39–0.92) (three studies, n=1225). The diagnostic accuracy of the GAD-2 questions in identifying anxiety disorders during early pregnancy remains unstudied. As women in early pregnancy are likely to have many anxieties (eg, over the viability of the pregnancy, decisional conflict over unplanned pregnancies), there may be high rates of 'false positives' when using the GAD-2 in pregnancy which could result in inappropriate referrals to mental health services.

We therefore aimed to investigate:

1. The UK prevalence of GAD, other anxiety disorders (including panic disorder, agoraphobia without panic disorder, social phobia, specific phobia, obsessive-compulsive disorder (OCD) and post-traumatic stress disorder (PTSD)) and comorbidity with other mental disorders during early pregnancy.
2. The sensitivity, specificity, positive predictive value (PPV), negative predictive value (NPV), positive likelihood ratio (LR+) and negative likelihood ratio (LR−) of the GAD-2 screening questions (on a Likert scale) compared with a gold standard diagnostic interview (Diagnostic and Statistical Manual of Mental Disorders, 4th edition (DSM-IV))[19] for identifying GAD, and for identifying any anxiety disorders (including panic disorder, agoraphobia without panic disorder, social phobia, specific phobia, OCD, PTSD and GAD) during early pregnancy. As DSM-V[20] no longer categorises OCD or PTSD as anxiety disorders, we also investigated diagnostic accuracy for any anxiety disorders by excluding OCD and PTSD.
3. How the sensitivity, specificity, PPV, NPV, LR+ and LR− change when the GAD-2 is scored and categorised as a yes (cut-off 1 or more) verses no (score of 0) response instead of the conventional cut-off of 3 or more (which could be asked by midwives at the same time as the two depression screening questions).

## METHODS
### Study design and participants
The WEll-being in pregNancy stuDY (WENDY study) was a cross-sectional survey that recruited women from an inner-city maternity service in South-east London using a sampling design stratified according to answering positive or negative (saying yes or no, respectively) on either of the two Whooley questions which are routinely asked by midwives as a mental health screen during the first antenatal booking appointment ("During the past month have you often been bothered by feeling down, depressed or hopeless?"; "During the past month have you often been bothered by having little interest or pleasure in doing things?"). A random sample of Whooley negative and all Whooley positive women were invited to participate.

Exclusion criteria were women aged under 16 years, women who declined to answer Whooley questions, those who had a termination or miscarriage prior to baseline interview or had already attended for their maternity booking appointment elsewhere in the UK. Eligible pregnant women who agreed to participate were recruited into the study as soon as possible after their first antenatal booking appointment, within a maximum of 3 weeks from the original booking appointment. Data collection of the index test (GAD-2 measure) and reference test (the gold standard diagnostic interview) were performed during the research interview, after written informed consent was obtained. Language interpreters were used where needed. For further details and power calculation of the WENDY study, see Howard et al.[21] Sample size of the current analysis was determined by the number of women with available data on the index test (GAD-2). Figure 1 shows flow chart of women through the WENDY study and those used for the current analysis.

### Research measures
GAD-2: This is a subscale of the Generalised Anxiety Disorder scale GAD-7 measure[22] and is a two-item self-report screen completed by women during the research interview. The questions include "Over the last 2 weeks, how often have you been bothered by any of the following problems? 1) Feeling nervous, anxious or on edge; 2) not being able to stop or control worrying". Answers are given on a Likert scale (not at all=0, several days=1, more than half the days=2 and nearly everyday=3). Scores range from 0 to 6, with a cut-off score of 3 or more indicative of anxiety symptoms.[23] The GAD-2 has also been used where YES to either question categorises an individual as a positive screen (used in clinical practice),[24] thus a score of 1 or more would indicate a possible YES answer. In this study, women's responses to the GAD-2 questions were categorised into the following groups:

1. GAD-2 (≥3): GAD-2 positive if they scored 3 or more (conventional scoring method), GAD-2 negatives if they scored <3.
2. GAD-2 (yes/no): GAD-2 positive if they scored 1 or more (indicating a 'yes' to either question), which is

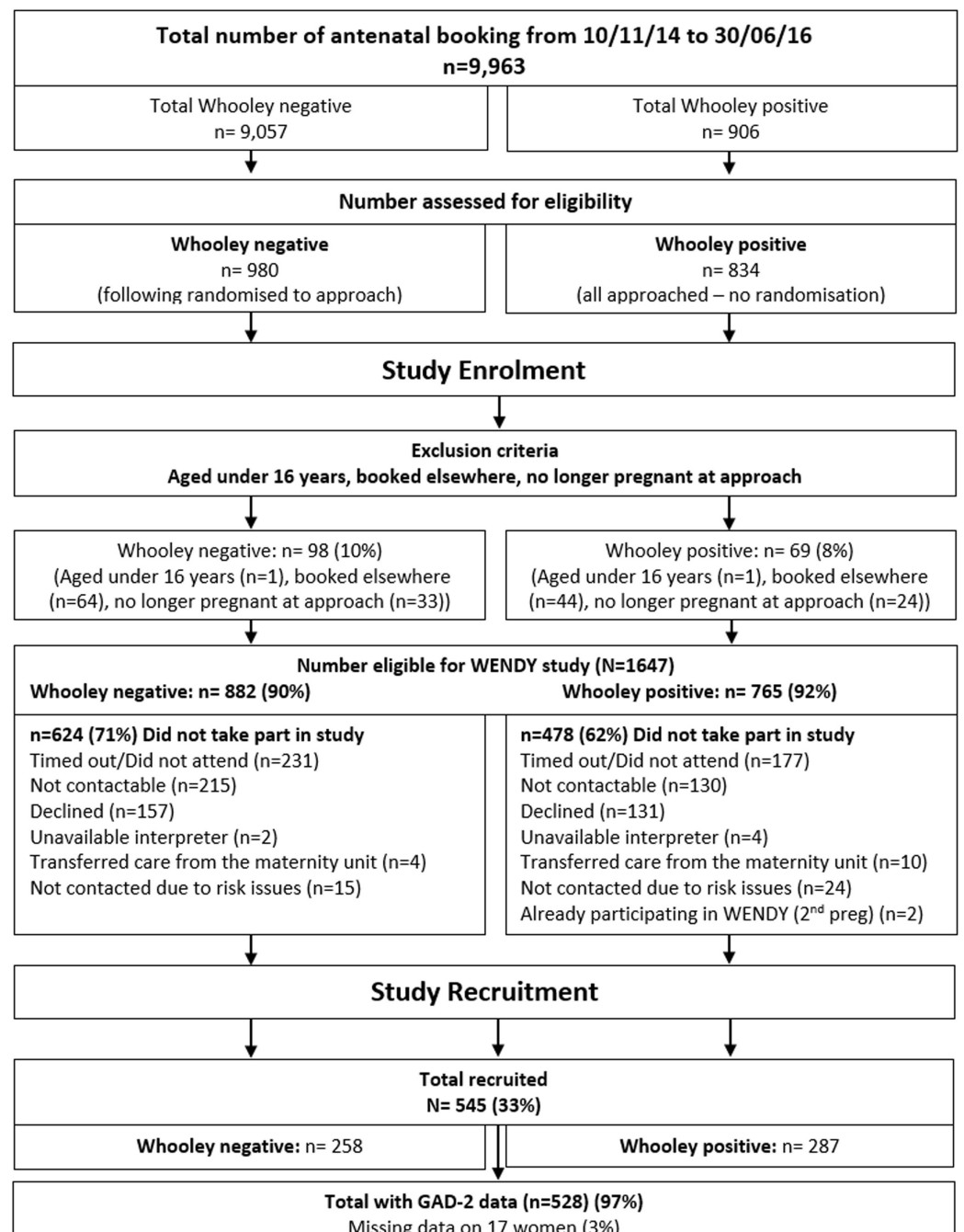

**Figure 1** Flow chart of women through the WEll-being in pregNancy stuDY (WENDY) study during the study period (total recruited n=545) and women with available data on the two-item Generalised Anxiety Disorder scale (GAD-2) (n=528).

used more in clinical practice, GAD-2 negative if scoring 0 (indicating 'no' to both questions).

### Anxiety disorders and comorbid disorders

The Structured Clinical Interview for DSM-IV Axis I Disorders (SCID-I): the SCID is a researcher-administered, semi-structured, and gold standard diagnostic interview consisting of standardised questions that correspond to each DSM-IV Axis I criteria.[19] We used the mood and anxiety disorders modules to generate diagnosis of depression, GAD, panic disorder, agoraphobia without panic disorder, social phobia, specific phobia, OCD and PTSD. Consensus on diagnosis was achieved during researcher's weekly meetings with LMH. Although clinical information and GAD-2 data were available at the time of these meetings, agreement on diagnosis was reached using responses given by women during the SCID interviews (ie, the GAD-2 items were not used to determine diagnosis of anxiety disorders).

As DSM-V[20] does not categorise OCD or PTSD as anxiety disorders and as the research version of the SCID for DSM-V was released after the start of the current study, we also carried out analyses for any anxiety disorders

according to those included in DSM-V as anxiety disorders, that is, excluding OCD and PTSD. We also used the eating disorder module to generate diagnosis for eating disorders including anorexia nervosa (including atypical), bulimia nervosa, binge eating disorder, purging disorder and other eating disorder.

## Patient involvement

The development of the WENDY study, outcome measures, grant application and study protocol were informed by our patient and carer advisory group. Meetings were held every few months to discuss the WENDY study and other related studies within a programme of work funded by the National Institute for Health Research (https://www.kcl.ac.uk/ioppn/depts/hspr/research/CEPH/wmh/projects/A-Z/esmi.aspx). The patient advisory group includes women with a range of mental disorders and have interest in our study programme. We circulated the results and draft manuscript (see 'Acknowledgements' section) to members of the group for comments. Patients were not involved in the recruitment or conduct of the study.

## Statistical analysis

Data were managed and analysed using Stata V.15.[25] Sampling weights were used to adjust for the oversampling of Whooley positive women[26] in all analyses apart from examining differences in sociodemographics between GAD-2 positives and GAD-2 negatives. The population prevalence of anxiety disorders and comorbid disorders were estimated based on weighted diagnostic interview responses (using Stata's 'svy' command). Bootstrap resampling of the weighted estimators was used for calculation of CIs.

As prespecified in our analysis plan, the weighted rates of 'true positive', 'false positives', 'true negatives' and 'false negatives' were tabulated for GAD, any anxiety disorder (including PTSD and OCD) and any anxiety disorder (without PTSD or OCD). Using these values, the sensitivity, specificity, PPV, NPV and LRs (positive and negative) were calculated.

### Missing data

Of the 545 participants, 24 (4%) women had some SCID missing data: one participant on GAD and all eating disorders: 1 participant on agoraphobia, specific phobia and PTSD; 1 participant on mixed anxiety and depression, borderline personality disorder and PTSD; 1 participant on hypomanic, manic, current major depressive disorder and bipolar I and II; 1 participant on all eating disorders and 19 participants on the PTSD module. List-wise deletion (performed in Stata) was used to calculate frequencies of SCID disorders in the study sample.

To calculate population prevalence of the SCID disorders, missing observations in the SCID items were accounted for by using inverse probability weights that incorporated the Whooley sampling, as well as variables that were significant in predicting missingness of SCID

responses (for full details of weightings and analysis strategy, see Howard *et al*[26]). Due to the large numbers of women who declined to respond to questions on the PTSD module, we carried out a sensitivity analysis for the prevalence estimate of PTSD in which we first assumed that all missing data were actually cases of PTSD, and then assumed that all missing data were not cases of PTSD.

Seventeen participants (3%) had missing GAD-2 data (15 for both questions, 1 for GAD-2 question 1 and 1 for GAD-2 question 2). No imputation was performed for women with missing data on GAD-2 items because missing data for one question would mean 50% data missing. These were therefore treated as missing observations and only women with complete data on both GAD-2 questions were used in the analyses to investigate sensitivity and specificity (n=528). Out of 528 women with complete data on the GAD-2 questions, there was missing data on the following SCID anxiety modules: 1 participant had missing data on PTSD, agoraphobia and specific phobia and 18 participants had missing data on the PTSD module.

## Ethics approval

The participants were provided with study information sheets, which was fully explained to them, had the opportunity to ask questions and gave informed consent prior to taking part in the study. No adverse events occurred for women taking part in the study during the research interview. Where risk (eg, significant suicidality, safeguarding issues) was identified during the research interview, researchers discussed this with the study PI and the woman's midwife and/or GP were informed, following consent to information sharing by the study participant (all women were aware this occurred and consented to this).

## RESULTS
### Sample characteristics

Between the dates of 10 November 2014 and 30 June 2016, 10 004 women attended their initial antenatal booking appointment with a midwife at the study site. Of these women, 41 did not have Whooley answers recorded so the base population consisted of 9963 women. The total number of eligible women recruited into the WENDY study was 545 women. This sample was similar to the base population on sociodemographic factors such as age, ethnicity and number of children.[21] Of the total number of women recruited into the WENDY study, 528 (97%) provided answers to the GAD-2 questions (figure 1). Sociodemographic characteristics (age, ethnicity and number of children) of women that provided GAD-2 answers were similar to the rest of the WENDY sample and wider base population (see online supplementary file 1). There were 119 (23%) GAD-2 (≥3) positives and 409 (77%) GAD-2 (<3) negatives within our study sample, where GAD-2 (≥3) positive was defined as reporting a total score of 3 or more on the GAD-2 questions (table 1). Compared with

**Table 1** Sociodemographics and characteristics of participants with GAD-2 data included in the diagnostic accuracy analyses (n=528)

| | GAD-2 negative score<3 n=409 | GAD-2 positive score≥3 n=119 | P values | Overall (total) n=528 |
|---|---|---|---|---|
| Age (years) | | | 0.126 | |
| 16–19 | 4 (1%) | 3 (3%) | | 7 (1%) |
| 20–29 | 104 (25%) | 40 (34%) | | 144 (27%) |
| 30–39 | 267 (65%) | 69 (58%) | | 336 (64%) |
| 40+ | 34 (8%) | 7 (6%) | | 41 (8%) |
| Ethnicity | | | 0.384 | |
| White | 223 (55%) | 56 (47%) | | 279 (53%) |
| Black/Caribbean | 123 (30%) | 46 (39%) | | 169 (32%) |
| Asian/Asian British | 19 (5%) | 3 (3%) | | 22 (4%) |
| Mixed/multiple ethnicity | 17 (4%) | 5 (4%) | | 22 (4%) |
| Other | 27 (7%) | 9 (8%) | | 36 (7%) |
| Highest education level | | | 0.006 | |
| None/school qualifications | 38 (9%) | 24 (20%) | | 62 (12%) |
| College/diploma/higher/certificate/training | 147 (36%) | 42 (35%) | | 189 (36%) |
| Degree level/postgraduate qualifications | 224 (55%) | 53 (45%) | | 277 (52%) |
| Employment status* | | | 0.010 | |
| Employed | 271 (66%) | 69 (58%) | | 340 (65%) |
| Student | 19 (5%) | 2 (2%) | | 21 (4%) |
| Unemployed | 42 (10%) | 19 (16%) | | 61 (12%) |
| Homemaker | 58 (14%) | 14 (12%) | | 72 (14%) |
| Not working due to illness/other | 18 (4%) | 14 (12%) | | 32 (6%) |
| Income† | | | <0.001 | |
| <£15 000 | 41 (13%) | 32 (35%) | | 73 (18%) |
| £15 000–£30 999 | 54 (17%) | 16 (17%) | | 70 (17%) |
| £31 000–£45 999 | 50 (16%) | 10 (11%) | | 60 (15%) |
| £46 000–£60 999 | 52 (16%) | 10 (11%) | | 62 (15%) |
| £61 000 or more | 119 (38%) | 24 (26%) | | 143 (35%) |
| Relationship status | | | <0.001 | |
| Single | 34 (8%) | 23 (19%) | | 57 (11%) |
| Partner but not cohabiting | 57 (14%) | 22 (18%) | | 79 (15%) |
| Married/cohabiting | 313 (77%) | 70 (59%) | | 383 (72%) |
| Separated/divorced | 5 (1%) | 4 (3%) | | 9 (2%) |
| Multiparous | | | | |
| No | 202 (49%) | 62 (52%) | 0.603 | 264 (50%) |
| Yes | 207 (51%) | 57 (48%) | | 264 (50%) |
| Planned pregnancy | | | | |
| Yes | 280 (68%) | 67 (56%) | 0.014 | 347 (66%) |
| Late booker | | | | |
| Yes | 344 (84%) | 94 (79%) | 0.191 | 438 (83%) |
| Translator required | | | | |
| Yes | 25 (6%) | 13 (11%) | 0.074 | 38 (7%) |

*Two participants had missing data on employment status (1 GAD-2 positive and 1 GAD-2 negative).
†One hundred twenty participants had missing data on income (93 GAD-2 negatives, 27 GAD-2 positives).

GAD-2 negatives, GAD-2 positives were more likely to be single, have lower levels of education, be unemployed or not working due to illness, have a lower income and have an unplanned pregnancy. When scoring for GAD-2 positives was defined as scoring 1 or more, there were 226 (43%) GAD-2 (yes/no) positives and 302 (57%) GAD-2 (yes/no) negatives within our study sample.

### SCID anxiety disorder prevalence and comorbidity
Using weighted estimates, the population prevalence was estimated as 17% (95% CI 12% to 21%) for any SCID anxiety disorder (or 15% (95% CI 11% to 19%) for any SCID anxiety disorder excluding PTSD and OCD, as in DSM-V). Specifically, there was an estimated population prevalence of 5% (95% CI 3% to 6%) for GAD, 4% (95% CI 2% to 6%) for social phobia, 8% (95% CI 5% to 11%) for specific phobia, 0.2% (95% CI 0.03% to 0.3%) for panic disorder, 0.4% (95% CI 0% to 2%) for agoraphobia, 2% (95% CI 1% to 4%) for OCD and 0.8% (95% CI 0% to 1%) for PTSD. As missing data were particularly common for the PTSD module, a sensitivity analysis demonstrated that when all women with missing data are assumed to have PTSD, the population prevalence estimate was 4% (95% CI 2% to 6%). When all women with missing data are assumed not to have PTSD, the population prevalence estimate was 0.8% (95% CI 0.1% to 1.4%). Of note, of the women who declined to respond to the PTSD questions, eight had already disclosed severe trauma earlier during the research interview which were related to being physically abused, sexual abuse/rape and witnessing violence.

The estimated population prevalence of comorbid depression with any anxiety disorder was 5% (95% CI 2% to 7%) (or 4% (95% CI 2% to 6%) for comorbid depression and any anxiety disorder excluding PTSD and OCD). Comorbid depression and GAD was estimated as 2% (95% CI 1% to 3%). Table 2 presents unweighted count (%) and weighted estimates of population prevalence (%) of comorbidity between SCID depression and all anxiety disorders (including PTSD and OCD).

### Diagnostic accuracy of the GAD-2 screening questions in identifying GAD
SCID GAD was found in 23 (6%) of GAD-2 (<3) negative and 35 (29%) of GAD-2 (≥3) positive women. After adjusting for weights, GAD was estimated to occur in 137 (2%, 95% CI 1% to 3%) of GAD-2 (<3) negatives and 302 (26%, 95% CI 15% to 40%) of GAD-2 (≥3) positives. SCID GAD was not found in 8439 (98%, 95% CI 97% to 99%) GAD-2 (<3) negatives and 872 (74%, 95% CI 60% to 85%) GAD-2 (≥3) positives (see online supplementary files 2 and 3 for cross-tabulation and proportions). Weighted sensitivity was 0.69, specificity 0.91, PPV 0.26, NPV 0.98, LR+ 7.35 and LR– 0.34. When scoring of the GAD-2 was changed to ≥1 (yes/no response), 58 (19%) of GAD-2 (yes/no) positive women met criteria for SCID GAD, whereas there were no GAD-2 (yes/no) negative women that met criteria for SCID GAD. After adjusting for weights, SCID GAD was estimated in 439 (11%, 95% CI 7% to 16%) of GAD-2 (yes/no) positives, whereas 5633 (100%) of GAD-2 (yes/no) negatives and 3678 (89%, 95% CI 84% to 93%) GAD-2 (yes/no) positives did not meet criteria for SCID GAD (see online supplementary files 2 and 4 for cross-tabulation and proportions). The weighted sensitivity was 1, specificity 0.60, PPV 0.11, NPV 1, LR+ 2.53 and LR– 0.

### Diagnostic accuracy of the GAD-2 screening questions in identifying any anxiety disorder
SCID anxiety disorders (including GAD, PTSD and OCD) were found in 73 (18%) of GAD-2 (<3) negatives and 57 (50%) of GAD-2 (≥3) positive women. After adjusting for weights, an estimated 1125 (13%, 95% CI 10% to 18%) of GAD-2 (<3) negatives and 404 (36%, 95% CI 23% to 51%) of GAD-2 (≥3) positives met criteria for any SCID anxiety disorder (including PTSD and OCD). SCID anxiety disorders were not found in 7268 (87%, 95% CI 82% to 90%) GAD-2 (<3) negatives and 723 (64%, 95% CI 49% to 77%) GAD-2 (≥3) positives (see online supplementary files 5 and 6 for cross-tabulation and proportions). Weighted sensitivity was 0.26, specificity 0.91, PPV 0.36, NPV 0.87, LR+ 2.92 and LR– 0.81. When scoring of the GAD-2 was changed to ≥1 (yes/no response), 22 (10%) of GAD-2 (yes/no) negative and 108 (37%) of GAD-2 (yes/no) positive women met criteria for any anxiety disorder (including PTSD and OCD). After adjusting for weights, 453 (8%, 95% CI 5% to 13%) of GAD-2 (yes/no) negatives and 1076 (27%, 95% CI 20% to 35%) of GAD-2 (yes/no) positives met criteria for any SCID anxiety disorder (including PTSD and OCD), whereas 5110 (92%, 95% CI 87% to 95%) GAD-2 (yes/no) negatives and 2881 (73%, 95% CI 65% to 80%) GAD-2 (yes/no) positives had no anxiety disorders (see online supplementary files 5 and 7 for cross-tabulation and proportions). Weighted sensitivity was 0.70, specificity 0.64, PPV 0.27, NPV 0.92, LR+ 1.95 and LR– 0.46. The exclusion of PTSD and OCD to the group of 'any' anxiety disorders made little difference (see table 3 and online supplementary file 8 for cross-tabulation). Sensitivity analysis of when PTSD missing data were considered as cases of PTSD and not cases of PTSD also did not make much difference (see table 3 and online supplementary file 9 for cross-tabulations).

### DISCUSSION
In this inner-city maternity population, the population prevalence estimated for GAD was 5% (95% CI 3% to 6%) and for all anxiety disorders was 17% (95% CI 12% to 21%), in line with other studies.[11] The population prevalence estimated for PTSD was lower (0.8%, 95% CI 0% to 1%) than the mean prevalence of 3.86% reported in a recent systematic review of PTSD identified by interviews during pregnancy.[27] However, as some women in our study who declined to answer the PTSD module also reported severe trauma elsewhere during the research

**Table 2** Comorbidities in the WENDY sample including unweighted count (%) and *weighted population prevalence estimates % (95% CI)*

| Disorder N | Depression* (n=545) | GAD (n=544) | Panic disorder (n=545) | Agoraphobia (n=544) | Social phobia (n=545) | Specific phobia (n=544) | PTSD (n=524) | OCD (n=545) | Eating disorders† (n=543) |
|---|---|---|---|---|---|---|---|---|---|
| Depression* (n=545) | – | *2% (1% to 3%)* | *0.2% (0% to 2%)* | *0.06% (0% to 0.3%)‡* | *1% (0% to 2%)* | *1% (0% to 3%)* | *0.4% (0% to 2%)* | *1% (0% to 2%)* | *0.2% (0% to 0.4%)‡* |
| GAD (n=544) | 39/544 (7%) | – | *0.13% (0.05% to 0.34%)‡* | *0.35% (0% to 2%)‡* | *1% (0% to 2%)* | *0.5% (0% to 1%)* | *0%* | *0.1% (0% to 0.3%)‡* | *0.2% (0% to 0.4%)‡* |
| Panic disorder (n=545) | 5/545 (1%) | 4/544 (1%) | – | *0%* | *0.03% (0% to 0.2%)‡* | *0%* | *0%* | *0.03% (0% to 0.2%)‡* | *0.03% (0% to 0.2%)‡* |
| Agoraphobia (n=544) | 2/544 (0.4%) | 1/543 (0.2%) | 0/544 (0%) | – | *0.4 (0.05% to 2%)‡* | *0%* | *0%* | *0%* | *0%* |
| Social phobia (n=545) | 9/545 (1.7%) | 6/544 (1%) | 1/545 (0.2%) | 1/544 (0.2%) | – | *1% (0% to 2%)* | *0.07% (0.02% to 0.3%)‡* | *0.5% (0% to 2%)* | *0.1% (0.05 to 0.4)‡* |
| Specific phobia (n=544) | 13/544 (2%) | 6/543 (1%) | 0/544 (0%) | 0/544 (0%) | 5/544 (1%) | – | *0%* | *1% (0.2% to 3%)‡* | *0.03% (0% to 0.2%)‡* |
| PTSD (n=524) | 11/524 (2%) | 0/523 (0%) | 0/524 (0%) | 0/524 (0%) | 2/524 (0.4%) | 0/524 (0%) | – | *0.03% (0% to 0.2%)‡* | *0%* |
| OCD (n=545) | 11/545 (2%) | 4/544 (1%) | 1/545 (0.2%) | 0/544 (0%) | 5/545 (1%) | 4/544 (1%) | 1/524 (0.2%) | – | *0.08% (0.02% to 0.3%)‡* |
| Eating disorders† (n=543) | 4/543 (1%) | 4/543 (1%) | 1/543 (0.2%) | 0/542 (0%) | 4/543 (1%) | 1/542 (0.2%) | 0/522 (0%) | 2/543 (0.4%) | – |

Italics indicates the *weighted population prevalence estimates % (95% CI)*.
*Depression includes all types of depression (major depressive disorder, mixed anxiety and depression).
†Eating disorders includes anorexia nervosa (including atypical), bulimia nervosa, binge eating disorder, purging disorder and other specified or eating disorder.
‡Bootstrap CI could not be performed due to small sample size. Weighted CI are presented which take into account sampling and missing SCID.
GAD, Generalised Anxiety Disorder; OCD, obsessive-compulsive disorder; PTSD, post-traumatic stress disorder; SCID, Structured Clinical Interview for DSM-IV Axis I Disorders; WENDY, WEll-being in pregNancy stuDY.

**Table 3** Sensitivity, specificity, PPV, NPV, LR+ and LR− of the GAD-2 questions (n=528)

| GAD-2 cut-off | GAD | Any anxiety disorders (including PTSD and OCD) | | | Any anxiety disorders (excluding PTSD and OCD)* |
| | | 19 missing data on PTSD (n=516)† | If missing cases considered as PTSD cases | If missing cases considered as not PTSD cases | |
|---|---|---|---|---|---|
| Cut-off≥3 | | | | | |
| Sensitivity | 0.69 | 0.26 | 0.26 | 0.26 | 0.26 |
| Specificity | 0.91 | 0.91 | 0.91 | 0.91 | 0.90 |
| PPV | 0.26 | 0.36 | 0.38 | 0.34 | 0.31 |
| NPV | 0.98 | 0.87 | 0.85 | 0.87 | 0.88 |
| LR+ | 7.35 | 2.92 | 2.84 | 2.82 | 2.71 |
| LR− | 0.34 | 0.81 | 0.82 | 0.81 | 0.82 |
| Cut-off≥1 (yes/no) | | | | | |
| Sensitivity | 1 | 0.70 | 0.70 | 0.70 | 0.68 |
| Specificity | 0.60 | 0.64 | 0.64 | 0.63 | 0.62 |
| PPV | 0.11 | 0.27 | 0.30 | 0.26 | 0.23 |
| NPV | 1 | 0.92 | 0.91 | 0.92 | 0.92 |
| LR+ | 2.53 | 1.95 | 1.95 | 1.90 | 1.81 |
| LR− | 0 | 0.46 | 0.47 | 0.47 | 0.51 |

*We excluded PTSD and OCD to present the difference, as DSM-5 no longer considered PTSD and OCD as anxiety disorders.
†Nineteen participants had missing data on the PTSD module. Seven of these participants met criteria for other anxiety disorders. Therefore, the total sample size for any anxiety disorders was 516.
DSM-5, Diagnostic and Statistical Manual of Mental Disorders, fifth edition; GAD, Generalised Anxiety Disorder; LR, likelihood ratio; NPV, negative predictive value; OCD, obsessive-compulsive disorder; PPV, positive predictive value; PTSD, post-traumatic stress disorder.

interview, we conducted a sensitivity analysis where missing data were assumed first to be cases of PTSD and then in a second analysis not to be PTSD cases: the prevalence estimate of PTSD was then potentially as high as 4% (95% CI 2% to 6%). Thus, the low prevalence of PTSD in our main analysis may reflect barriers to disclosure of trauma and associated symptoms.

As others have reported,[28 29] comorbidity is also common and we found estimates for comorbid depression and GAD (2%, 95% CI 1% to 3%) that were similar to previous estimates derived from representative samples and using diagnostic clinical interviews (GAD and depression 2% (95% CI 0% to 3%).[28] However, the population prevalence estimated in our sample of comorbid depression and any anxiety (5%, 95% CI 2% to 7%) was slightly lower than previous estimates (7%, 95% CI 3% to 11%).[28]

This study makes a novel contribution to the gap in the literature by formally examining the diagnostic accuracy of the GAD-2 screening questionnaire for women during early pregnancy. The GAD-2 (using a cut-off of ≥3; as recommended by NICE) had a reasonable LR when used in early pregnancy for identifying GAD (LR+ 7, ie, above 5, which has been suggested to indicate a potentially useful tool in clinical practice).[30] However, GAD is not very common and the PPV is low (26%). Furthermore, the diagnostic accuracy of the GAD-2 for identifying other anxiety disorders was poor (LR+ 2.92). This evidence suggests that the GAD-2

is not a helpful tool for maternity services as it will generate many false positives.

There are several strengths to this study. First, we recruited a stratified representative sample of women using language interpreters to facilitate inclusion of non-English-speaking women.[21] Second, we used an efficient and robust sampling design which facilitated the estimation of population prevalences and we used a gold standard diagnostic interview. Limitations include recruitment from a single maternity site in London, although the single South London maternity site used in this study included a very ethnically and socioeconomically diverse population. There were some missing data (although this was rare other than for PTSD), and timing of administering the GAD-2 in early pregnancy may have overinflated the risk of 'false positives'.

### Implications

This study does not support the NICE recommendation to use the GAD-2 in early pregnancy, due to its low PPV even for GAD (when applying a cut-off of both ≥3 or ≥1 indicating a yes response) and low effectiveness for 'any anxiety disorder'. Its accuracy later in pregnancy warrants further study but a recent systematic review reported the prevalence of self-reported anxiety symptoms to increase through pregnancy (trimester 1: 18%, trimester 2: 19%, trimester 3: 25%)[11]; diagnostic accuracy is therefore unlikely to improve.

Following the NICE CG192 guideline recommendations, some services have already implemented the GAD-2 in routine practice. The findings from the current paper challenge the use of expert consensus and extrapolation from evidence derived from the general population to pregnant women when making the NICE recommendations specifically for pregnant women. We argue that recommendations for pregnant women should be evidence based and propose that, at present, only the NICE recommendation on use of the Whooley questions is supported by evidence.[21] Currently, the evidence suggests that implementation of routine use of the GAD-2 is unwarranted. This brings us to a number of potential unexplored future research directions, which include a comprehensive diagnostic accuracy study, including data on acceptability, of the full GAD-7 questionnaire in pregnancy and the three 'anxiety' items in the Edinburgh Postnatal Depression Scale in pregnancy. For the GAD-7, currently there is no evidence. There is some conflicting evidence for the three 'anxiety items' in the Edinburgh Postnatal Depression Scale, but as others have suggested, research is needed to determine their validity, reliability and diagnostic accuracy as a measure of antenatal anxiety disorders.[31 32] Furthermore, use of other routinely collected data including the Whooley questions could be used in a model (using predictive modelling techniques) to identify anxiety disorders or perhaps even more importantly, 'any mental disorder'. Finally, further research is needed to examine ways to overcome the barriers to disclosure of trauma and trauma-related symptoms in pregnancy. This suggests that more sensitive methods need to be developed to ask women about trauma.

## CONCLUSIONS

This study suggests that anxiety disorders are common in early pregnancy, but the GAD-2 screening measure is not useful during early pregnancy. In addition, even if asked directly, PTSD may be particularly difficult to detect in routine practice (as some women may decline to answer specific questions in relation to symptoms arising from traumatic experiences) and therefore further research is needed on how to overcome barriers to disclosure of trauma-related symptoms .

**Author affiliations**
¹Section of Women's Mental Health, Health Service and Population Research Department, Institute of Psychiatry, Psychology & Neuroscience, King's College London, London, UK
²Biostatistics and Health Informatics Department, Institute of Psychiatry, Psychology and Neuroscience, King's College London, London, UK
³Warwick Clinical Trials Unit, Warwick Medical School, University of Warwick, Coventry, UK
⁴Department of Women and Children's Health, School of Life Course Sciences, Faculty of Life Sciences and Medicine, King's College London, London, UK
⁵Women's Health, King's College Hospital NHS Foundation Trust, London, UK
⁶Parent-Infant Research Institute (PIRI), Austin Health, Melbourne School of Psychological Sciences, University of Melbourne, Melbourne, Victoria, Australia
⁷South London and Maudsley NHS Foundation Trust, London, UK

**Acknowledgements** The authors are grateful for the advice received regarding the WENDY study from the Patient and Public Advisory Group, the Programme Steering Committee (Professor Rona McCandlish (Chair), Dr Heather O'Mahen, Dr Pauline Slade, Ceri Rose and Rosemary Jones) and the Data Monitoring and Ethics Committee (Roch Cantwell (Chair), Liz McDonald-Clifford, Marian Knight, Stephen Bremner). The authors also want to take the opportunity to thank the women who participated in this study.

**Contributors** LMH, DB, JM and AP conceived, planned and obtained funding for the ESMI programme of research and they, with KT, wrote the first draft of the WENDY study protocol. JD was involved in modifying study protocol. SN was involved in the literature search, planning and carrying out statistical analysis, interpretation of data and drafted the manuscript. EGR and AP provided guidance on statistical analysis. Both LH and EGR were involved with interpretation of data and revision on manuscript drafts. All authors commented on subsequent drafts, interpretation of data and approved final version of manuscript. Professor LMH (senior author and chief investigator) is the guarantor of the study.

**Funding** This paper summarises independent research funded by the National Institute for Health Research (NIHR) under the Programme Grants for Applied Research programme (ESMI Programme: grant reference number RP-PG-1210-12002) and the NIHR/Wellcome Trust King's Clinical Research Facility and the NIHR Biomedical Research Centre and Dementia Unit at South London and Maudsley NHS Foundation Trust and King's College London. The senior author (LMH) also has salary support from an NIHR Research Professorship (NIHR-RP-R3-12-011).

**Disclaimer** The views expressed are those of the author(s) and not necessarily those of the NHS, the NIHR or the Department of Health. The study team acknowledges the study delivery support given by the South London Clinical Research Network. The funder had no role in the design or conduct of the study; the collection, management, analysis and interpretation of the data; or the writing of the article.

**Competing interests** Louise M Howard (senior author and chief investigator), chaired the National Institute for Health and Care Excellence CG192 guidelines development group on antenatal and postnatal mental health in 2012-14. No other conflicts of interest.

**Patient consent** Obtained.

**Ethics approval** The research was approved by the National Research Ethics Service, London Committee—Camberwell St Giles (ref no 14/LO/0075).

**Provenance and peer review** Not commissioned; externally peer reviewed.

**Data sharing statement** Full study protocol (approved by Research Ethics Committee) and patient level data is available from Chief Investigator Professor Louise Howard (louise.howard@kcl.ac.uk). Consent for data sharing was not obtained but the presented data are anonymised and risk of identification is very low.

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
