## [Reviewer comments · BMJ Open]

ARTICLE DETAILS

TITLE (PROVISIONAL)	The prevalence and identification of anxiety disorders in pregnancy: the diagnostic accuracy of the two item Generalized Anxiety Disorder scale (GAD-2)
AUTHORS	Nath, Selina; Ryan, Elizabeth; Trevillion, Kylee; Bick, Debra; Demilew, Jill; Milgrom, Jeannette; Pickles, Andrew; Howard, Louise

VERSION 1 – REVIEW

REVIEWER	Bronwyn M Graham University of New South Wales Sydney Australia
REVIEW RETURNED	16-May-2018

GENERAL COMMENTS	This manuscript addresses a critical question- whether the GAD-2 is an accurate screening devise to identify the presence of antenatal anxiety, as currently recommended by NICE. The rationale for the study, the relevant background information, and the hypothesis regarding the potential lack of accuracy of the GAD-2 (with respect to concerns regarding excessively high false positives) were clearly outlined in the introduction. Additionally, given the dearth of research on the prevalence of anxiety disorders in the antenatal period (relative to research on antenatal depression), this study addresses a clear need to acquire current and accurate statistics in this area. The methods are sound, with the inclusion of the multiple means of scoring the GAD-2 of potential practical benefit. The results were analysed appropriately and cautiously. The interpretations were in line with the results and the limitations of the study were fairly appraised. Overall, the manuscript was extremely well written and the results are of strong practical importance. I have no suggestions for improvement; this is an extremely strong manuscript.
---

REVIEWER	Hamideh Bayrampour UBC, Canada
REVIEW RETURNED	23-May-2018

GENERAL COMMENTS	Thank you for an opportunity to review this manuscript. This is a very well-written paper on a timely topic. Findings have important implications for research and practice. Please see below for some comments: -results, p.10, please consider reporting weighted estimate numbers without decimals (e.g., 302 vs. 302.2) as the latter is not meaningful. - discussion. p.12, the authors noted that "However, in our study, when missing data were assumed to indicate cases of PTSD, the prevalence estimate was 4% (95%CI: 2 to 6%), highlighting the possibility of barriers to disclosure of trauma and associated symptoms." Why did the authors make the assumption that all women with missing data would have PTSD. Additionally linking
--

	missing data to disclosure barriers may be seen as an "over-interpretation" unless the authors provide more rational. Similarly conclusion point in the conclusion section should be revised.
--	---

VERSION 1 – AUTHOR RESPONSE

Reviewer: 1

There were no specific issues to address. We thank the reviewer for the very positive comments.

Reviewer: 2

Thank you for an opportunity to review this manuscript. This is a very well-written paper on a timely topic. Findings have important implications for research and practice. Please see below for some comments:

Thank you for the positive comments. We have addressed all requests below.

-results, p.10, please consider reporting weighted estimate numbers without decimals (e.g., 302 vs. 302.2) as the latter is not meaningful.

Thank you. We now report weighted estimate numbers without decimals (changes made on pages 10 & 11).

- discussion. p.12, the authors noted that "However, in our study, when missing data were assumed to indicate cases of PTSD, the prevalence estimate was 4% (95%CI: 2 to 6%), highlighting the possibility of barriers to disclosure of trauma and associated symptoms." Why did the authors make the assumption that all women with missing data would have PTSD. Additionally linking missing data to disclosure barriers may be seen as an "over-interpretation" unless the authors provide more rational. Similarly conclusion point in the conclusion section should be revised.

The population prevalence estimate for PTSD in our study was lower (0.8%, 95%CI: 0 to 1%) than the mean prevalence (3.86%) reported in previous studies. Therefore, our intention was to examine this further and then to suggest a possible explanation for the lower prevalence of PTSD found in our study in the context of a sensitivity analysis. We now make this clearer in the discussion by stating: "However, as some women in our study who declined to answer the PTSD module also reported severe trauma elsewhere during the research interview, we conducted a sensitivity analysis where missing data were assumed firstly to be cases of PTSD and then in a second analysis not to be PTSD cases: the prevalence estimate of PTSD was then potentially as high as 4% (95%CI: 2 to 6%). Thus, the low prevalence of PTSD in our main analysis may reflect barriers to disclosure of trauma and associated symptoms." (page 12, paragraph 1).

As suggested by the reviewer, we have also revised the conclusion as follows:

"In addition, even if asked directly, PTSD may be particularly difficult to detect in routine practice (as some women may decline to answer specific questions in relation to symptoms arising from traumatic experiences) and therefore further research is needed on how to overcome barriers to disclosure of trauma related symptoms." (page 14, paragraph 1).

FORMATTING AMENDMENTS FROM EDITORIAL OFFICE:

- Please re-upload each figure under 'Image' file designation with at least 300 dpi resolution and at least 90mm x 90mm of width.

We have re-uploaded the following figures according to the above specifications:

Figure 1

Online supplementary file 3

Online supplementary file 4

Online supplementary file 6

Online supplementary file 7

- Please include figure legends at the end of your main manuscript.

We have added more text to figure 1 legend to provide more detail about the figure and also uploaded separately under 'Image' as suggested above. As this comment says to include at the end of the main manuscript, we have also put it at the end of the manuscript (page 19) and editorial team can use the version that is most suitable for publication.

VERSION 2 – REVIEW

REVIEWER	Hamideh Bayrampour UBC canada
REVIEW RETURNED	16-Jul-2018
GENERAL COMMENTS	I have no further comments.